# FoveaTer: Foveated Transformer for Image Classification

## Abstract

Many animals and humans process the visual field with varying spatial resolution (foveated vision) and use peripheral processing to make eye movements and point the fovea to acquire high-resolution information about objects of interest. This architecture results in computationally efficient rapid scene exploration. Recent progress in vision Transformers has brought about new alternatives to the traditionally convolution-reliant computer vision systems. However, the Transformer models do not explicitly model the foveated properties of the visual system nor the interaction between eye movements and the classification task. We propose foveated Transformer (FoveaTer) model, which uses pooling regions and eye movements to perform object classification tasks using a vision Transformer architecture. Our proposed model pools the image features using squared pooling regions, an approximation to the biologically-inspired foveated architecture, and uses the pooled features as an input to a Transformer Network. It decides on subsequent fixation locations based on the attention assigned by the Transformer to various locations from previous and present fixations. The model uses a confidence threshold to stop scene exploration, dynamically allocating more fixation/computational resources to more challenging images. After reaching the stopping criterion, the model makes the final object category decision. We construct a Foveated model using our proposed approach and compare it against a Full-resolution model, which does not contain any pooling. On the ImageNet-100 dataset, our Foveated model achieves the accuracy of the full-resolution model using only 35% transformer computations and 73% overall computations. Finally, we demonstrate our model's robustness against adversarial attacks, where it outperforms the full-resolution model.

## 1 Introduction

Many mammals, including humans, have evolved a locus (the fovea) in the visual sensory array with increased spatial fidelity and use head and eye movements (Land, 2012; Marshall et al., 2014) to orient such locus to regions and objects of interest. The system design allows visual-sensing organisms to accomplish two objectives: fast target detection crucial for survival and savings in the computational cost. Computational savings are accomplished by limiting the number of units with high computational costs (i.e., higher spatial resolution processing) to the fovea's small spatial region. Fast target detection is achieved by distributing the remaining computational power across a much larger area in the periphery, with a lower spatial resolution with increasing distance from the fovea. Critical to the design is an efficient algorithm to guide through eye movements the high-resolution fovea to regions of interest using the low-resolution periphery (Hayhoe & Ballard, 2005; Strasburger et al., 2011; Ludwig et al., 2014) and allow optimizing the target detection and scene classification. Various computational models were proposed to model the search using foveated visual system (Yamamoto et al., 1996; Prince et al., 2005).

Computer vision has evolved from using hand-crafted features to data-driven features in modern CNNs. Due to their computational limitations, the objectives of the computer vision systems align well with those of human visual system: to optimize visual detection and recognition with an efficient computational and metabolic footprint. Approaches towards saving computational power can be seen; for example, computer vision systems evolved from using sliding windows to RCNN's (Gir-

shick et al., 2014) use of selective search and Faster-RCNN's (Ren et al., 2015) use of Region Proposal Network (RPN).

A system that mimics the human vision by processing the scene with a foveated system and rational eye movements has also been proposed. This approach to exploring the scene can be seen in models like RAM (Mnih et al., 2014) for recognizing handwritten single-digits or detecting objects (Akbas & Eckstein, 2017) where they sequentially process the image and decide what to process next by using the peripheral information. These foveated models approach that of full-resolution models but using a fraction of the computations. Foveated systems have also shown to result in more robustness (Luo et al., 2015; Deza et al., 2019; Deza & Konkle, 2020; Kiritani & Ono, 2020; Vuyyuru et al., 2020) against adversarial attacks.

There has been a recent innovation in computer vision of using Transformers (Touvron et al., 2020; Dosovitskiy et al., 2020) for object classification tasks which departs from the traditional over-reliance on convolutions. Even after replacing the convolutions with attention modules and multilayer perceptrons, visual Transformers (Dosovitskiy et al., 2020; Touvron et al., 2020) achieve close to state-of-the-art performance on the ImageNet dataset and provide better robustness against adversarial attacks (Shao et al., 2021).

Due to the flattened architecture of the transformers, it is easier for multi-resolution features to share the same feature channels. Transformers (Vaswani et al., 2017) have the added benefit of self-attention , which facilitates the interaction of various parts of the image irrespective of distance. No papers have evaluated the additional potential gains of incorporating a foveated architecture into Vision transformers for the task of ImageNet classification.

Here, we evaluate the effect of a foveated architecture and sequential eye movements on a state of the art transformer model sitting on a convolutional backbone. We compare the foveated transformer relative to the full-resolution model in terms of classification accuracy and robustness to adversarial attacks. Furthermore, we extend previous papers by investigating the separate contributions to robustness related to the foveated processing (Luo et al., 2015; Deza et al., 2019; Deza & Konkle, 2020; Kiritani & Ono, 2020; Vuyyuru et al., 2020) and that of the process of fixating images at different locations.

To this end, we perform an object classification task using multiple fixations, moving foveal attention across different parts of the image, and using only a limited portion of the image information at each fixation, thereby reducing the input to the transformer by many folds. The model decides on subsequent fixation locations using the learned self-attention weights and fixation locations of all the fixations until the current step. Finally, the model integrates information across fixations using average-pooling to make the final classification decision.

## 2 RELATED WORK

**Transformers** have achieved great success in the field of Natural Language Processing since their introduction by Vaswani et al. (2017) for machine translation. Recently, the application of Transformer models in computer vision has seen tremendous success. Vision Transformer (ViT) model introduced by Dosovitskiy et al. (2020) achieved remarkable performance on ImageNet (Deng et al., 2009) by using additional data from JFT 300M (Sun et al., 2017) private dataset. Subsequently, the DeiT model (Touvron et al., 2020) introduced knowledge transfer concepts in transformers to leverage the learning from existing models. Using augmentation and knowledge transfer, the DeiT model achieved close to state-of-the-art performance using training data from the ImageNet dataset alone.

**Sequential processing** provides three main advantages in computer vision. Larochelle & Hinton (2010) proposed a model based on the Boltzmann machine that uses foveal glimpses and can make eye movements. First, it can limit the amount of information to be processed at a given instant to be constant, i.e., the ability to keep computations constant irrespective of the input image size. Second, sequential models can help model human eye movement strategies and help transfer that information to build better computer vision systems. RAM (Mnih et al., 2014) introduced a sequential model capable of making a sequence of movements across the image to integrate information before classification. In addition, the hard-attention mechanism, implemented using reinforcement learning, was used to predict the sequence fixation locations. Ba and Minh (Ba et al., 2015) extended these ideas to recognize multiple objects in the images on a dataset constructed using MNIST. Third, se-

quential processing requires fewer parameters than a model using full-resolution image input. Other models (Xu et al., 2015) have proposed image captioning models based on both hard-attention and soft-attention. Additionally, the spatial bias introduced into CNNs due to padding (Alsallakh et al., 2021) can be overcome using sequential models (Tsotsos, 2011)

**Computational models of categorization and eye movements** have been proposed for rapid categorization in terms of low-level properties such as spatial envelopes (Oliva & Torralba, 2001) and texture summary statistics (Rosenholtz et al., 2012). Saliency-based models (Koch & Ullman, 1987; Itti et al., 1998; Itti & Koch, 2000) traditionally tried to model eye movements by identifying bottom-up properties in the image that will capture attention. Torralba et.al (Torralba et al., 2006) showed how saliency could be combined with contextual information to guide eye movements. Low-resolution periphery and high-resolution central fields are integrated with saliency to predict human-like eye-movements (Wloka et al., 2018). Akbas (Akbas & Eckstein, 2017) implemented a biologically-inspired foveated architecture (Freeman & Simoncelli, 2011) with a deformable parts model to build a foveated object detector which accuracy was close to a full-resolution model but using a fraction of the computations. Spatial transformer networks (Jaderberg et al., 2015) were used with foveation to improve object localization using foveated convolutions (Harris et al., 2019) and achieve better eccentricity performance (Dabane et al., 2021).

**FoveaTer** combines an approximation to biologically-inspired foveated architecture with a Vision Transformer Network. We apply our model to real-world images from the ImageNet dataset for the task of image classification. We pool the input to the transformer using the pooling architecture described in the following sections based on the fixation location, which reduces the number of inputs to the transformer. The subsequent fixation location is given by a confidence map, constructed using the attention weights from the last transformer block. Feature vectors corresponding to the class token from different fixations are averaged using an average-pooling layer followed by a fully connected layer, resulting in the final classification decision.

A novel aspect of the proposed work is that the model also learns that all images are not equally difficult to classify adapting the exploration eye movements to different image classes and thus varying computational resources used to classify different images successfully. The model implements this idea using a confidence threshold to restrict the scene exploration to the necessary fixations to classify the image.

Also novel is an evaluation the adversarial robustness of our model to understand the separate contributions of the foveated architecture and that of sequential fixations towards defense against adversarial attacks. We use the fast gradient sign method (FGSM (Goodfellow et al., 2014)), which is an adversarial attack that computes the adversarial image by backpropagating the gradients once, and the projected gradient descent method (Kurakin et al., 2017; Madry et al., 2018), which iteratively computes the adversarial image, as our adversarial attacks.

## 3 MODEL

**DeiT-Tiny (Touvron et al., 2020):** The DeiT-Tiny architecture begins with a convolution embedding layer that transforms the $[3, 224, 224]$ input image into a $[192, 14, 14]$ representation that is input into a series of twelve transformer blocks, each sized for a 192-dimensional embedding.

**Full-resolution model:** Our Full-resolution model refers to a hybrid adaptation, transformer network with convolutional backbone, of the DeiT-Tiny architecture using the same number of parameters. This hybrid model replaces the single convolutional layer of the DeiT-Tiny model with a convolutional backbone. The convolutional backbone is composed of first two stages of the ResNet-18 (He et al., 2016) architecture, which transforms the input from $[3, 224, 224]$ to $[128, 28, 28]$, followed by three convolutional layers to acheive a final feature map of size $[192, 14, 14]$. The first convolutional layer transforms the input feature dimensionality from 128 to 192 using kernels of size 3 and stride 1, resulting in $[192, 28, 28]$. The second convolutional layer downscales the features using a kernel of size 3 and a stride of 2, resulting in features of size $[192, 14, 14]$. The third convolutional layer uses a kernel size of 1 and stride of 1, retaining the size of the feature map. The number of transformer blocks is reduced from 12 to 9 to match the number of parameters with the DeiT-Tiny model.

**Foveated model:** Extending the hybrid architecture described above, our Foveated model analyzes a pooled-version of the features from the convolution backbone at each time step and arrives at the final category decision after making the required number of time steps to satisfy the decision criterion. We show the architecture of our model in Figure 1. The input image is first passed through the convolutional backbone described above, resulting in size $[192, 14, 14]$. Then, we perform fixation dependent average-pooling, using the "Foveation module" (see next section), on the features from the convolutional backbone resulting in features of size $[192, 29]$, where the fixation refers to one of the 196 possible spatial locations in the input feature map. Under this non-uniform average-pooling model, locations closer to the fixation location use smaller neighborhoods for pooling than locations far from the fixation location. Pooled features of size $[192, 29]$ along with the class token are passed through the nine transformer blocks, where class token refers to a learnable vector of size 192 values. Since the transformer blocks retain the dimensionality of the input, the output of the transformer blocks has the same size of $[192, 30]$. We use the self-attention weights, corresponding to the class token feature vector from the last transformer block, in the "Accumulator module" (see next section) to predict the following fixation location. Feature vector corresponding to the class token is given as input to the average-pooling layer, averaging across such vectors received from all the fixations. Finally, the classification layer transforms the average-pooled feature vector into a logits vector.

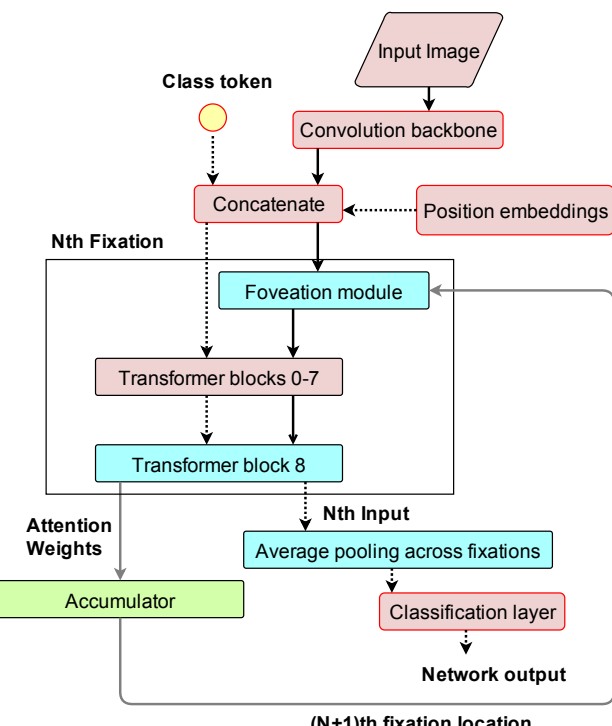

Figure 1: **FoveaTer architecture:** Solid black arrows denote the flow of image-related features. The foveation module performs fixation-dependent pooling, on the full-resolution features ($[192, 14, 14]$) output of the convolution backbone, resulting in pooled feature vectors ($[192, 29]$). During training, a random location in the image serves as the initial fixation. *Accumulator* uses the attention weights from the last transformer block to predict the next fixation location. The Average-pooling layer takes the feature vector corresponding to the class token as input from each past fixation, followed by a final classification layer.

### 3.1 FOVEATION MODULE

Inspired from the radial-polar pooling regions proposed in Freeman & Simoncelli (2011), we use squared pooling regions in this paper for computational speed-up. Each image in a mini-batch has a corresponding fixation location, where the fixation location represents the center of the visual field shown in Figure 2, thereby allowing us to align the input image/feature map with the visual field. After aligning the input feature map with the visual field, features falling within a pooling region are

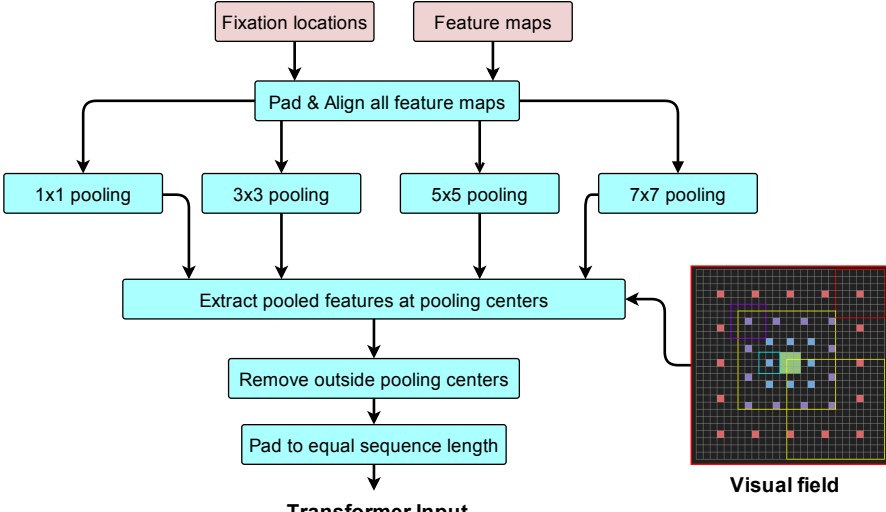

Figure 2: **Foveation module:** Inputs to the module are aligned to the visual field and are zero-padded to have the same size as the visual field. We pool the input features according to the pre-defined pooling neighborhoods in the visual field, resulting in one feature vector for each of the 49 pooling regions in the visual field, colored blocks in the visual field figure correspond to the centers of these 49 pooling regions. After ignoring the pooling centers falling in the padded region instead of on the original image, the remaining pooled features are zero-padded to a constant sequence length of 29.

average-pooled, and the resultant pooled vector represents that pooling region. Highlighted regions in Figure 2 correspond to the centers of the pooling regions, 49 blocks.

We use pooling regions with receptive field sizes $1 \times 1$, $3 \times 3$, $5 \times 5$, $7 \times 7$ blocks on a feature map of size $14 \times 14$ blocks, which is the output of the convolution backbone. Each block corresponds to a $[16, 16]$ pixel region in the input image of width and height 224. The visual field is of size $27 \times 27$ blocks. Central green $3 \times 3$ block represents the high-resolution Fovea, where there is no average-pooling. The next ring of pooling regions, where the center of the pooling region is blue, has a receptive field of $3 \times 3$ which translates to average-pooling of 9 feature vectors to generate the representative feature vector for that pooling region. Similarly, the rings of violet and red-colored pooling centers have receptive fields of $5 \times 5$ and $7 \times 7$ blocks, respectively. Yellow squares are of size $14 \times 14$ blocks, i.e., equal to the feature map of the image, thereby illustrating the size of the image size to the visual field, which is approximately one-quarter.

We ignore the pooling regions falling outside the image, and the output of the foveation module is zero-padded to maintain the same length across all images present in a batch since different fixation locations give rise to an unequal number of active pooling regions. We use 29 as the feature length as it is the maximum possible number of active pooling regions. The feature vector corresponding to each pooling region is computed as follows,

$$P_i = (1/M) \sum_{j=0}^{M-1} E_{ij} \tag{1}$$

Where $i = \{1, \ldots, 49\}$ corresponds to the pooling region index, $E_{ij}$ is the embedded feature vector present in that pooling region, $M = \{1, 9, 25, 49\}$ corresponds to number of features inside the pooling region and $P_i$ is the mean feature vector corresponding to that pooling region.

## 3.2 GUIDED FIXATION SEQUENCE PREDICTION:

Self-attention weights from the last transformer block are collected to get each pooling region's contribution to the feature vector corresponding to the class token. A confidence map is constructed based on the fixation point location by putting these weights back on a $14 \times 14$ map at the corresponding pooling region's location, where $14 \times 14$ corresponds to the size of the input feature map. After each fixation, we degrade the attention weights in the confidence map at a rate of 50%. Inhibition of return (IOR) (Dukewich & Klein, 2015) refers to a tendency in human observers not to attend at previously attended or fixated regions. We constructed an IOR map by applying an IOR of $3 \times 3$ blocks around a fixation location and degraded it by 50% after each fixation. After subtracting the

IOR map from the confidence map, the following fixation location is extracted based on the location of the maximum in the confidence map.

**Loss function:** We use Cross-entropy for computing the classification loss. Loss from all fixations is incorporated to get the mini-batch loss, as shown below.

$$loss = \sum_{i=1}^{N} \sum_{j} y_j \log(p(y_j)) + (1 - y_j) \log(1 - p(y_j)) \qquad (2)$$

Where $i$ corresponds to the fixation index, $N = 5$ for the Foveated model & $N = 1$ for Full-resolution model, i.e., single-pass, $y$ corresponds to the target label, $j$ corresponds to the class index, and $p$ correspond to the predicted probability.

## 4 RESULTS

We train the model weights from scratch and no pre-trainng. The Foveated model is trained for five fixations to have a computational complexity lower than the Full-resolution model. In the following sub-sections, we compare the performance and computational complexity of the Foveated model performance against the Full-resolution model. In addition, we compare the adversarial robustness of the Foveated and Full-resolution models. Appendix 6.1 consists of ablation studies regarding architecture choice, contribution of the attention mechanism, and the importance of peripheral rings.

### 4.1 IMPLEMENTATION DETAILS

**Training:** We train all the 5.5M parameters of the model. Using initial learning of $1e - 5$ and a minimum learning rate of $1e - 6$, we train both Foveated and Full-resolution models for 300 epochs. We use AdamW (Kingma & Ba, 2014; Loshchilov & Hutter, 2019) optimizer with a decay of $1e-8$ and a cosine learning rate schedule. We use the hyperparameters and augmentation settings from DeiT implementation (Touvron et al., 2020). We use ImageNet-100 (Tian et al., 2020) and ImageNet (Deng et al., 2009) datasets for the results shown in the following sub-sections. For ImageNet-100, We use the train validation split provided by the ImageNet and report results on the validation set. We use GeForce GTX 1080 Ti for training and testing purposes. Both Foveated and Full-resolution models take 25 hrs and 21 hrs respectively to train for 300 epochs on three GPUs. The Foveated model made five fixations on each input image during training, with the initial fixation starting at a random location. Due to our model's flexibility, the same model can make any desired number of fixations.

**Dynamic-stop of Fixation Exploration:** Due to various factors such as occlusion, camera angle, brightness, the difficulty of making a classification decision varies across images. To achieve higher computational efficiency in our Foveated model during inference, we stop exploring the images with fixations when the predicted class with the highest probability reaches a pre-defined threshold corresponding to that class. The threshold equals the mean probability of all the correct predictions belonging to that class in the training dataset.

**Images/Sec:** We report the number of inferences completed by the GPU during a one-second time interval to compare the computational complexity of different models during inference time.

**Initial fixation for the Foveated model:** The input feature map to the *Foveation module* has a spatial size of $14 \times 14$, and thus consider any one of these 196 fixation locations. During training, a random location is selected as the initial fixation point, and the model guides subsequent fixations. During inference, for the results presented in Table 1 and Table 2, the initial fixation is set at the center of the image to remove any variation in results arising from the randomness in the initial fixation location.

### 4.2 TOP-1 ACCURACY:

The full-resolution model allows the long-range full-resolution interactions between the feature vectors, whereas the Foveation model serially explores the image allowing only a limited number of interactions between pooled feature vectors at each given fixation. Using the guided fixation prediction method described in the previous section, the Foveated model makes four guided fixations after

Table 1: **Performance on ImageNet100 (Tian et al., 2020) dataset:** We find that the Foveated model outperforms the Full-resolution model both in terms of performance and computational complexity. Pure-Trans refers to the model with a single convolutional layer, and Hybrid refers to the models with convolutional backbone at the start of the transformer as described in the previous sections. For evaluation, the initial fixation for the Foveated model is chosen at the center of the image to remove effects due to variations of the initial fixation location.

| Model | Type | Params | Top-1 | Conv Flops | Trans Flops | Total Flops |
|---|---|---|---|---|---|---|
| Deit-Tiny | Pure-Trans | 5.5M | 76.3 | 0.029 | 1.046 | 1.07 |
| Full-resolution (**Ours**) | Hybrid | 5.3M | 84.58 | 1.245 | 0.697 | 2.03 |
| Foveated (**Ours**) | Hybrid | 5.3M | 84.66 | 1.245 | 0.531 | 1.84 |
| Foveated with Dynamic-Stop (**Ours**) | Hybrid | 5.3M | **84.68** | 1.245 | **0.245** | 1.49 |

Table 2: **Performance on ImageNet dataset:** We compare our models against the state-of-the-art models with a similar number of parameters using three metrics: **Top-1 accuracy**, **Image throughput**, and **Adversarial robustness**. For reporting adversarial robustness, $\epsilon$ is set at $0.01$, and the adversarial attack is computed for the transformer front-end only; therefore, we do not report comparisons to the convolutional models. **DS:**Dynamic-Stop. Throughput for the dynamic-stop model is computed as a weighted combination of throughputs of the Foveated model with a constant number of fixations, i.e., 1 to 5 fixations. DeiT-Ti (distilled) refers to the transformer architecture aided by a convolution network.

| Model | Type | Params | Images/sec | Top-1 | Top-1-Adv ($\epsilon = 0.01$) | |
|---|---|---|---|---|---|---|
| | | | | | FGSM | PGD |
| ResNet-18 (He et al., 2016) | Conv | 11.7M | 1505 | 69.8 | — | — |
| EfficientNet-B0 (Tan & Le, 2019) | Conv | 5M | 720 | 77.1 | — | — |
| Deit-Ti (Touvron et al., 2020) | Pure-Trans | 5.7M | 796 | 72.2 | 46.53 | 36.5 |
| Deit-Ti (distilled) (Touvron et al., 2020) | Trans-ConvT | 6M | 796 | 74.5 | — | — |
| Full-resolution (**Ours**) | Hybrid | 5.5M | 652 | 73.6 | 39.37 | 28.98 |
| Foveated (**Ours**) | Hybrid | 5.5M | 637 | 70.2 | **59.24** | **57.67** |
| Foveated-DS (**Ours**) | Hybrid | 5.5M | 960 | 69.9 | — | — |

the initial random fixation. Sample fixations made by the model are shown in Figure 3a and average number of fixations made by the model per class are shown in Figure 3b. Additional visualization of what is visible to the model at each fixation is shown in Appendix 6.2.

### 4.2.1 RESULTS ON IMAGENET-100

Due to the hybrid nature of the models proposed, incorporating a CNN backbone and a transformer network, the Foveated and Full-resolution models outperform the DeiT-Tiny model, as shown in Table 1. The Foveated model's accuracy is slightly above the Full-resolution model but critically has lower computational cost. Critically, the last row shows that with the addition of the dynamic-stop, the FoveaTer model can achieve close to 65% computational savings in transformer flops and 26% savings in terms of overall flops while retaining the performance achieved by the Full-resolution model.

### 4.2.2 RESULTS ON IMAGENET

We present the results on the ImageNet dataset in Table 2. The Foveated model achieves an accuracy above the ResNet-18 model. For the dynamic-stop, we first compute the number of images falling in each of the five fixation bins and throughputs of the Foveated models with one to five fixations. The throughput of the Dynamic-stop model is computed as the weighted average of the throughputs of individual fixation models using the bin count as the weights. Using the dynamic-stop, the throughput of the the Foveated model is improved (637 → 960), as shown in the last row. Appendix 6.3 shows the correlation between object complexity and the number of fixations made by the Dynamic-stop model.

### 4.3 ROBUSTNESS AGAINST ADVERSARIAL ATTACKS

We demonstrate the robustness of our foveated system in this section. We consider the two adversarial attacks, Fast Gradient Sign Method (FGSM Goodfellow et al. (2014)) and Projected Gradient

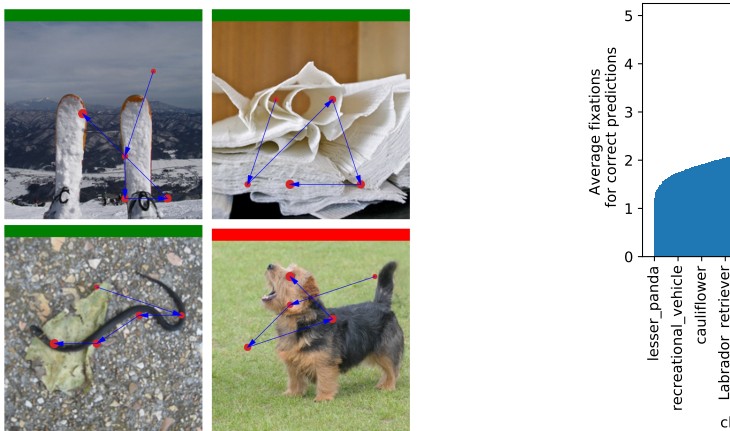

(a) **Fixation sequences:** A maximum of five fixations are made for each image with Dynamic-stop. The initial random fixation is followed by four guided fixations. Each subsequent fixation center is represented with a bigger red circle. Blue arrows represent the transition from one fixation to the next. Green and Red bars on top correspond to correct and wrong predictions respectively.

(b) **Average fixations per class:** Average number of fixations used by each class are plotted on y-axis. Some of the class names are displayed on the x-axis.

Figure 3: Fixation sequences and Average fixations per class

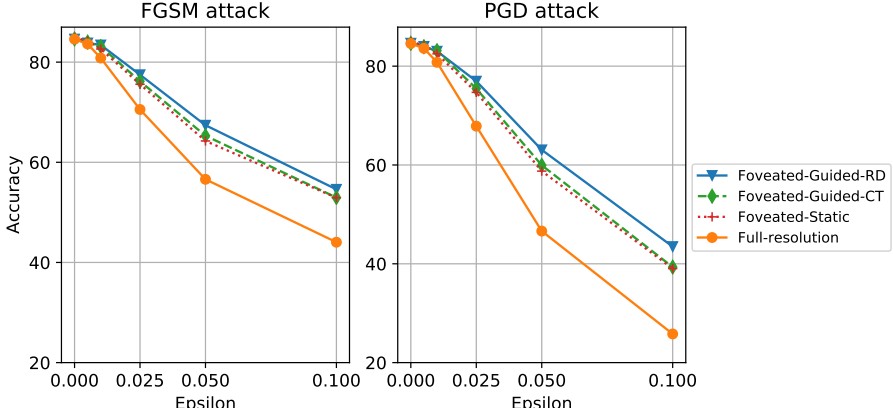

Figure 4: **Adversarial robustness:** Comparison of Foveated model (with guided fixations and two different initial fixations, *RD* - initial random fixation. *CT* - initial fixation at the image center), a foveated model that makes the same fixation sequence for every image (Foveated model with static fixations), and the Full-resolution model. Epsilon represents the strength of the attack. As the strength of the attack increases the accuracy of the Foveated model with guided fixations accuracy is greater than the Foveated model with static fixations while the Full-resolution model has the lowest accuracy. Raw values are available in Table 5 in Appendix.

Descent (PGD Kurakin et al. (2017); Madry et al. (2018)), to compare the robustness of Foveated and Full-resolution models. We use Cleverhans library (Papernot et al., 2018) for implementing the adversarial attacks.

To compute the adversarial robustness of the model, we separated the network into a convolutional backbone and a Transformer front-end. The adversarial attack is computed only for the Transformer front-end, i.e., features from the convolutional backbone as network input and Transformer front-end as the model. Since our model only affects the Transformer part of the network, this disassociates the interaction of the adversarial attack with the convolutional backend, which does not interact with the foveation module. For this reason, purely convolutional networks are not considered. We compare only the DeiT-Tiny model with Full-resolution and Foveated models. Epsilon corresponds to the strength of the attack.

Overall, Foveated model outperforms both DeiT-Tiny and Full-resolution models in terms of adversarial robustness.

### 4.3.1 RESULTS ON IMAGENET-100

Figure 4 shows that performance of the Full-resolution model (84.58) and Foveated model (84.62) are similar for zero epsilon value, but as the strength of the attack increases, the Foveated model outperforms the Full-resolution model. The robustness of the Foveated model can be either due to the pooling operations that are part of the model or our model's inherent ability to dynamically decide fixation paths, thereby minimizing the effect of the attack by automatically following a different path from that of the adversarial model.

To isolate effect of foveation from the influence of fixation exploration, We compare various configurations of the Foveated model. For the foveated guided fixations model, we explored a version with a random initial fixation and another version with a constant initial fixation at the center of the image. The model then dynamically selected the subsequent four fixations, thereby allowing the possibility of different fixation paths for the adversarial image computation and model evaluation. The model with the random initial fixation results in more variability in subsequent fixations. Critically, we also used a pre-defined fixation sequence (center initial fixation followed by four fixations at the center of each quadrant), thereby ensuring that the adversarial image computation and the model evaluation that follows will use the same fixation locations. We refer to this model as the Foveated static.

Results in Figure 4, show that the Foveated guided model with an initial random fixation outperforms the static fixation model, suggesting that the variability in fixation sequences contributes towards adversarial robustness. The comparison to the Foveated Guided model with the fixed initial fixation at the center of the image shows that a stochastic component to the initiation can be important to vary the fixation sequence and result in greater robustness.

Finally, we illustrate the contributions of the foveated nature of the model by comparing the Foveated model with the five pre-defined fixations (*red curve, Foveated static*) to the Full-resolution model (*orange curve*). This pre-defined fixation path Foveated model will also be the desirable solution when the computational time of a serial fixation model is a bottleneck, as all the pre-defined fixations can run in parallel. Figure 4, show that the Foveated model with static fixations outperforms the Full-resolution model illustrating the contributions of the foveated nature(Luo et al., 2015; Kiritani & Ono, 2020; Vuyyuru et al., 2020) of the model to robustness to adversarial attacks.

### 4.3.2 RESULTS ON IMAGENET

We show the results of adversarial attacks in the last two columns of Table 2. Due to architectural changes from the DeiT-Tiny model, the Full-resolution model (*FGSM*-39.4*; PGD*-29) exhibits lower adversarial robustness compared to Deit-Tiny model (*FGSM*-46.5*; PGD*-36.5). On the other hand, Foveated model (*FGSM*-59.2*; PGD*-57.7), which shares architectural similarity with the Full-resolution model, exhibits higher adversarial robustness than both DeiT-Tiny and Full-resolution architectures.

## 5 CONCLUSION

We provided a comprehensive framework for using foveal processing and fixation exploration on a vision transformer architecture for image classification. The proposed architecture introduces a way to limit computations required to process an image by flexibly adjusting the required number of fixations, allowing larger batch sizes during training, providing robustness to adversarial attacks, and giving us a model that can allocate computational resources based on the difficulty of an image. Our findings reveal that variability in fixation sequences to images can contribute over foveated processing to robustness to adversarial attaches. In conclusion, we leveraged the most recent Vision Transformer architecture and combined it with ideas from foveated vision to come up with a model which has multiple knobs in terms of the number of fixations to be made and limits on the computations done so that the end-user will have the flexibility to fine-tune depending on their needs.

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

Table 3: Performance of the ImageNet pre-trained models on the auxiliary task without any additional training.

| Pre-trained Model | Top-1 |
|---|---|
| ResNet-18 | 1.3 |
| EfficientNet-B0 | 1.8 |
| Deit-Ti | 7.8 |
| Full-resolution (**Ours**) | 10.9 |
| Foveated (**Ours**) | 7.5 |

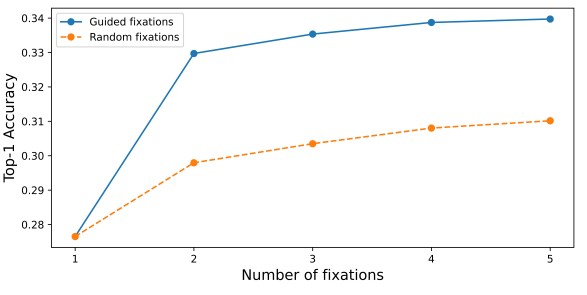 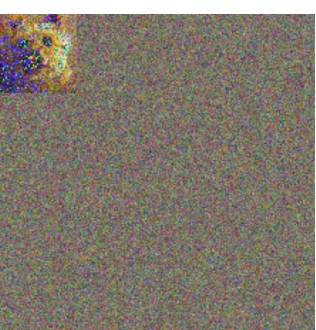

(a) Comparison between random fixations and guided fixations with crop located at top-left corner and the initial fixation at bottom-right corner.

(b) Sample evaluation image

Figure 5: Additional task to determine the effectiveness of guided fixations

# 6 APPENDIX

## 6.1 ABLATION STUDIES

### 6.1.1 ABLATION STUDY 1: ATTENTION MECHANISM

Objects in the ImageNet dataset occupy a high percentage of the image area resulting in a diminished difference between guided and random fixations. In order to illustrate the performance of the attention mechanism, we create this small extension task.

**Task**: $224 \times 224$ image from the ImageNet dataset is down-scaled to a size of $56 \times 56$ and is randomly placed in $224 \times 224$ image with normal random noise ($\mathcal{N}(\mu, \sigma^2)$, where $\mu = 0$ and $\sigma = 0.5$). Table 3 shows the performance of the ImageNet pre-trained models on this new task. Transformer-based models, including our hybrid models, easily outperform the CNN models. Sample image is shown in Figure 5b.

**Training:** Model trained on ImageNet dataset is fine-tuned for 30 epochs with a lower learning rate of $5e - 5$ on this auxiliary task. Initial fixation is at a random location during training.

**Evaluation:** Chance value is $0.1$ due to 1000 classes. The down-scaled image and the initial fixation are at the top-left corner and bottom-right corner, respectively, to start at the maximum possible distance from the target for both guided fixations and random fixations conditions.

**Results:** Comparison of the model with guided fixations is plotted against random fixation model in Figure 5a. The guided model consists of one initial fixation at the bottom-right, followed by four guided fixations. The random fixation model consists of one initial fixation at the bottom-right followed by four random fixations. The difference in performance illustrates the contributions of the fixation prediction component of our model.

### 6.1.2 ABLATION STUDY 2: INPUT FEATURES TO FOVEATION MODULE

Table 2 consists of a hybrid version of the Foveated model where a convolutional backbone provides input features to the foveation module. To assess the contribution of this convolution backbone, we

Table 4

| Model | Type | Params | Top-1 |
|---|---|---|---|
| Foveated **(Ours)** | Hybrid | 5.5M | 70.2 |
| Deit-Ti (Touvron et al., 2020) | Pure-Trans | 5.7M | 72.2 |
| Foveated-T **(Ours)** | Pure-Trans | 5.7M | 68.5 |

replaced the convolution backbone with transformer layers, making the resultant pure-transformer model (Foveated-T) entirely equivalent for the DeiT-Ti model except for the presence of foveation module. In this model, the Foveation module is preceded by four transformer layers followed by eight transformer layers, where the original DeiT-Ti model consists of Twelve transformer layers. Results are shown in Table 4. The hybrid version of the Foveated model has a Top-1 accuracy of 70.2 compared to 68.5 for the pure-Transformer version of the Foveated model. Therefore, the presence of the CNN backbone contributes to an improvement of 1.7%.

### 6.1.3 Ablation study 3: Contribution of peripheral features

As described in the Foveation module section, Fovea is $3 \times 3$ followed by three peripheral rings of pooled features. In this section, we look at the contribution of those peripheral rings towards the final performance. We reuse the task described in Section 6.1.1 to get the contribution of those peripheral features. To this end, we train four different models on this auxiliary task. Foveated-F consists of only Fovea and no additional peripheral features. Foveated-F3 consists of Fovea and one peripheral ring of $3 \times 3$ average pooled peripheral features. Foveated-F35 consists of Fovea and two peripheral rings, $3 \times 3$ and $5 \times 5$ average pooled peripheral features. Foveated-F357 consists of Fovea and three peripheral rings, $3 \times 3$, $5 \times 5$, and $7 \times 7$ average pooled peripheral features. Foveated-F357 is equivalent to the model presented in Section 6.1.1.

All four models start with the weights of the Foveated model trained on ImageNet. Then the model is fine-tuned for 30 epochs. Figure 6 compares the performance of all four models under random fixations condition described in Section 6.1.1. Models with more peripheral rings outperform the models with lower peripheral rings, and the models converge to saturation performance with an increase in the number of fixations.

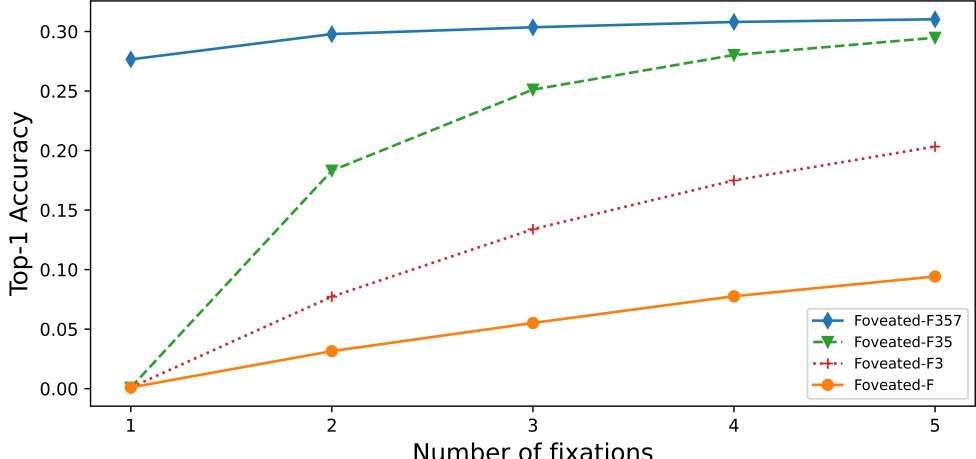

Figure 6: **Model Comparison:** Down-sized images are present at the top-left corner, and initial fixation is set at the bottom-right corner. Models with more periphery rings outperform the models with less number of periphery rings.

## 6.2 VISUALIZATION:

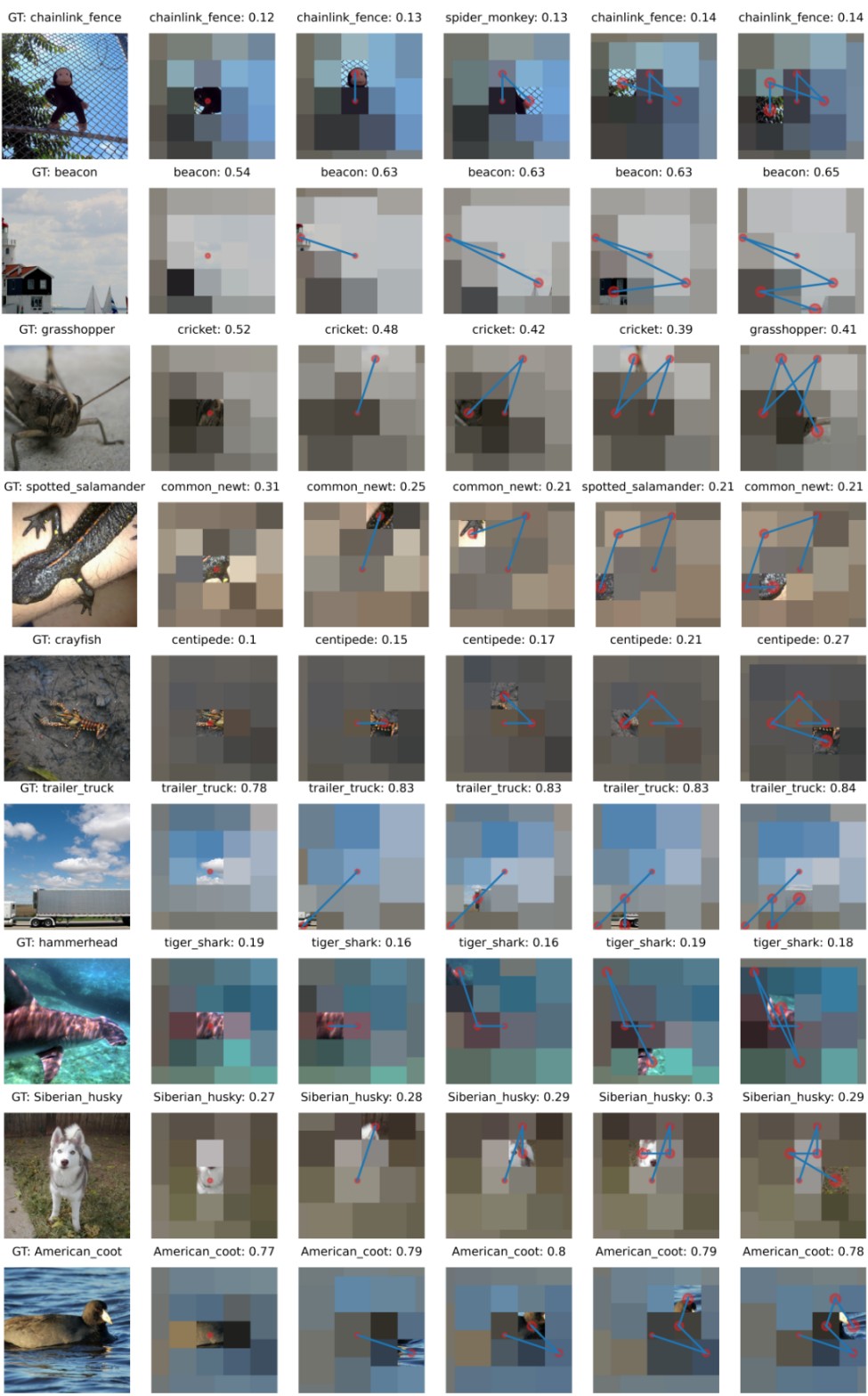

Figure 7: **What the model sees:** Each row contains five images that correspond to the five fixations on a single image. Left-most is the original image with the ground-truth (GT) class.

Table 5: Adversarial robustness on ImageNet100: dataset. **Foveated-S-RD:** Foveated model with guided fixations and random initial fixation. **Foveated-P:** Foveated model with pre-defined fixations with parallel execution capability. **Foveated-S-CT:** Foveated model with guided fixations and image center as initial fixation.

| | DeiT-Tiny | | Full-resolution | | Foveated-S-RD | | Foveated-P | | Foveated-S-CT | |
|---|---|---|---|---|---|---|---|---|---|---|
| Epsilon ($\epsilon$) | FGSM | PGD | FGSM | PGD | FGSM | PGD | FGSM | PGD | FGSM | PGD |
| 0 | 76.34 | 76.34 | 84.58 | 84.58 | 84.60 | 84.64 | 84.94 | 84.94 | 84.62 | 84.62 |
| 0.005 | 69.82 | 69.26 | 83.62 | 83.60 | 83.82 | 83.98 | 84.22 | 84.22 | 84.0 | 83.94 |
| 0.01 | 58.62 | 55.10 | 80.82 | 80.78 | 83.42 | 83.02 | 82.62 | 82.50 | 83.12 | 83.14 |
| 0.025 | 35.90 | 23.74 | 70.54 | 67.86 | 77.48 | 76.96 | 75.58 | 74.70 | 76.18 | 75.46 |
| 00.05 | 23.56 | 10.14 | 56.6 | 46.62 | 67.40 | 63.06 | 64.26 | 58.76 | 65.30 | 59.94 |
| 0.1 | 15.42 | 6.58 | 44.04 | 25.80 | 54.60 | 43.48 | 52.92 | 39.04 | 52.96 | 39.34 |
| 0.2 | 10.24 | 5.56 | 35.08 | 8.38 | 46.26 | 25.18 | 44.80 | 21.46 | 44.54 | 21.28 |

## 6.3 CORRELATION OF OBJECT COMPLEXITY AND NUMBER OF FIXATIONS:

Using the Foveated model trained on ImageNet, we compute an entropy metric for each image to quantify the object complexity. Using a $4 \times 4$ grid, the foveated model was run for one fixation at these 16 locations resulting in 16 probability values per image for the ground-truth class. The entropy of the image is computed using these values. For each of the thousand classes, the correlation between the number of fixations made using Dynamic-Stop and the entropy of those images is calculated. Results are shown in Figure 8. Out of the 1000 classes, 815 classes have a negative Pearson-correlation coefficient. When the object is small and occupies a small percentage of image area, the resulting entropy will be low as the one-fixation model will not detect the target everywhere on the uniform grid. At the same time, when the object is small, the Dynamic-Stop model will need more fixations to detect the target as it can not make the right decision by fixating anywhere on the image. A negative correlation from the plot captures the inverse relationship of entropy and the required number of fixations.

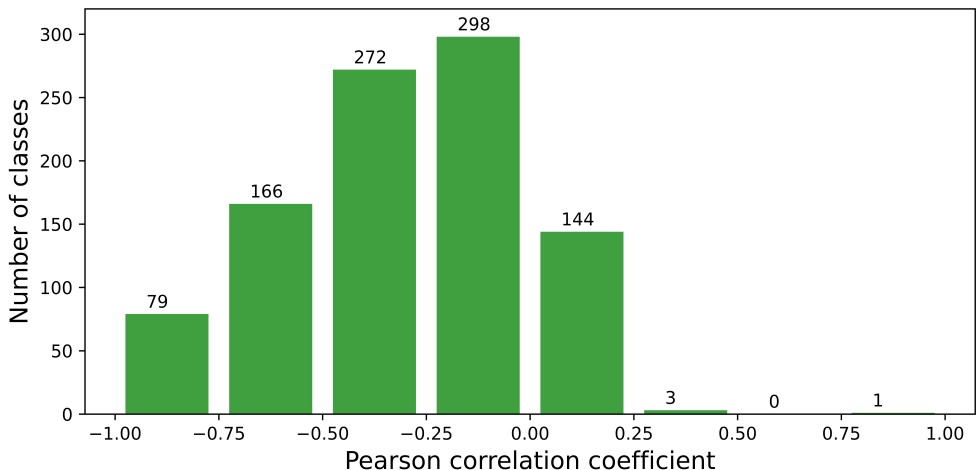

Figure 8: Relationship between image entropy and the number of fixations made by the Foveated model with Dynamic-stop. Pearson correlation coefficient computed for each class separately, and the histogram displays the distribution of correlation coefficient value across classes.

