# OpenReview forum: "FoveaTer: Foveated Transformer for Image Classification"
_ICLR.cc/2022/Conference — ICLR 2022 Submitted_

### Official Review · Reviewer_gFeo · 2021-11-01

**Correctness:** 2
**Technical Novelty And Significance:** 3
**Empirical Novelty And Significance:** 2
**Recommendation:** 5
**Confidence:** 4

**Main Review:**

While recent success in convolutional processing has dominated the field, certain aspects of vision common to many biological vision systems have had limited treatment. The proposed work is interesting in this respect, as it combines a foveated vision model within the popular transformer model framework. It demonstrates a model for active vision with predicted eye movements using recurrent network to predict the next area of foveated attention as a simulated eye movement.

The introduction provides motivation for foveated vision and review of prior art in this area. However, the most relevant works are not cited and the paper falsely claims to be the first for combining transformer networks and foveated vision: "No papers have evaluated the additional potential gains of incorporating a foveated architecture into transfomers." The authors should correct this claim, and take note of the following references to explain what is unique in their approach and how it differs from other foveated vision models:

* Harris, Ethan William Albert, Mahesan Niranjan, and Jonathon Hare. "Foveated convolutions: improving spatial transformer networks by modelling the retina." (2019).
* Dabane, Ghassan, Laurent Perrinet, and Emmanuel Daucé. "What You See Is What You Transform: Foveated Spatial Transformers as a bio-inspired attention mechanism." (2021).

The authors reference 2017 work of Akbas and Eckstein as another foveated object detector, but do not provide distinctives that set their model apart from the work. The current paper would benefit from direct comparison to these other foveated vision methods. The only comparisons to other methods are against full-resolution models.

It is unclear from the introduction why adversarial attacks are a focus of this paper. The motivation of the paper starts on the basis of biologically inspired vision systems, yet such systems do not have to address adversarial attack other than overcoming natural camouflage (which is not at all similar to the types of adversarial attacks compared in the paper involving gradient informed adversarial examples), and in any case there is not explanation of why adversarial robustness should be a feature of foveated vision, or why to evaluate foveated vision systems on this basis.

Similarly, while the connection between transformer networks and foveated vision systems seems more supportable, it is lacking. There is an obvious opportunity and need to provide some description of transformer networks and why they are particularly well suited a framework to support the foveated vision system the authors wish to demonstrate.

Eye movements seem limited by a 14x14 grid. Explanation of the foveation pooling and aggregation approach was a little confusing. Is the movement restricted to an inner region, or is the zero-padding extend to potentially half the visual field when the attention is placed near the edge of the image. It would be interesting to visualize the visual field for a given fixation. Given the biological motivation, I would be interesting to know how this compares to biological systems.

Classification results seem a little underwhelming. While it barely beats ResNet-18, 70.2% vs 77.1% against EfficientNet-B0 is disappointing. Again, without direct comparisons to other foveated vision systems it is difficult to tell how much this improves (if at all) on current art.

I was particularly intrigued by what could be learned from the saccades themselves, but relatively little was discussed in that regard. The examples in Fig 3 were helpful to see but limited to a few hand-picked examples, and difficult to know how well movements relate to object content. It would be informative to compare eye movements predicted by the model to biological movements in gaze-tracking datasets.
Such comparisons could support the biological motivation of the paper and show that the model is learning beneficial movements. Comparing the only to full resolution does not sufficiently answer this. Another comparison to illustrate this would be to compare the existing model against random movements.

I think thresholding the classification probability is a good idea to determine a stopping point for the dynamic stopping of fixation exploration, and the average fixations per class is interesting but I would like to understand if this correlates in some way to actual object complexity. The conclusion claims the learned model can allocate computational resources based on the difficulty of an image, but I did not see any comparisons that defended this. Can you show that more fixations were used by the model for images that were more complex by some entropy or other complexity metric? Or compare to human gaze experiments?

Other specific comments:
Pg. 2: "reach that approaches" -> "approach"
Pg. 2: "using which locations far apart in the image will interact" -> this sentence is unclear as to its meaning, but needs revision
Pg. 4: "squared pooling regions...for computational speed-up": what kind of speedup is obtained, and by how much, and compared to what?
Pg. 6: description of dynamic-stop of fixation exploration: Is this only for inference, or is dynamic stopping performed during training?
Figure 4: Requires color to interpret. It may be helpful to use different marker and line styles in case this is viewed in black and white





**Summary Of The Paper:**

The authors present an approach to computer vision combining Transformer networks and foveated vision. They contrast their approach, which uses a recurrent attention model to simulate eye saccades to explore a scene, against full resolution deep CNN models for classification problems to demonstrate reduced computation and greater resistance to adversarial attacks.

**Summary Of The Review:**

This is an interesting area of research, but I do not think that the authors did a sufficient job of motivating the model and experiments. They did not motivate the use of transformer networks, or explain why adversarial attacks are an appropriate problem for comparing foveated vision systems. They also left some important questions answered or not fully answered (such as whether the fixation movements were actually benefiting the classification). Experiments provided do not do direct comparison with equivalent models with and without the predictive attention which could easily determine the contribution for this part of the model. Claims in the paper are not sufficiently supported. A priority claim made in the paper as the first work to combine foveated vision and transformer networks is false and ignores published work by Harris et al. 2019 that uses spatial transformer networks for foveated convolutions. Direct comparisons to other foveated vision systems are lacking, and without these it is not possible to assess how well their particular approach compares to the most similar prior art.

---

> ### Author Response · Authors · 2021-11-23
> **Response to reviewer gFeo**
>
> Thanks for your review.
>
> - Regarding the comparison to Harris et al. 2019 and Dabane et al. 2021,
>
> Our work uses the vision transformers ([1], [2]), whereas those studies use a spatial transformer. These are different techniques. Self-attention is the key component of the vision transformer architectures. In contrast, the spatial transformer is an attention mechanism that helps the network be invariant to inputs scale and rotation. To clarify this point, we updated the statement in the paper to reflect that we use the vision transformer. The previous studies are applied to MNIST and CIFAR-10 datasets, whereas we present a model to work with the ImageNet dataset. We added the two papers to the references.
>
> - Regarding the 2017 work of Akbas and Eckstein,
>
> That study processes images at multiple scales while we do not. Pooled features are only used as input to the classifier, and no part of the network exclusively operates on the pooled features except for the classification layer. It was limited to 20 classes of the PASCAL dataset compared to our results on the entire ImageNet dataset.  Their model is based on the older deformable parts model ([3]). They make the eye movements to locations with the highest evidence for the target after classifying all the templates, whereas we use an attention mechanism to direct eye movements before the final classification. Additionally, unlike that study, we investigate the adversarial robustness of our model.
>
> - Regarding adversarial robustness in the context of foveated vision,
>
> Adversarial attacks can fool Deep Networks by making image changes that are barely detectable to humans. For this reason, robustness against adversarial attacks is a critical property for artificial vision systems that have become important as part of evaluations ([4], [5], [6]).
>
> - Regarding the usefulness of transformer-based networks for foveated vision systems,
>
> When foveation is applied on a feature map, the 2D structure of the input feature map is lost due to different sized pooling regions. If convolution layers follow this foveation, it is difficult to learn the convolution kernels on irregularly spaced pooling regions. As transformers are convolution-free, they provide the flexibility to operate on features from different pooling regions making them more practical for foveated artificial vision systems.
> Fixations are restricted to be within the image. As the size of the grid is 14x14, it has 196 possible locations. Attention map has a value for each of the pooling regions.
>
>
> - Regarding visualization,
>
> please refer to Appendix 6.2. Please note that the foveation module operates on features of length 192 instead of input images; therefore, the visualization generated on the input image is an approximation by looking at the corresponding receptive fields for each pooling region.
>
>
> - Regarding the comparison with other foveated systems, since they were implemented in much smaller datasets and more specific tasks, the direct comparison is difficult. However, we added a pure-transformer version of the Foveated model in Appendix 6.1.2
>
>
> - Regarding the comparison of Guided movements to Random movements,
>
> please refer to the Appendix 6.1.
>
>
> - Regarding the correlation of object complexity with the number of fixations,
>
> please refer to the Appendix 6.3.
>
>
>
>
>
> - The speed-up refers to the log-polar pooling regions ([7]) for which no standard pooling functions are available in deep learning libraries, unlike the average-pooling operations on square regions, resulting in more computational time. Dynamic-Stop was used only in the inference time.
>
> We addressed the other minor comments in the text as well.
>
> - References:
> 1. Dosovitskiy, Alexey, Lucas Beyer, Alexander Kolesnikov, Dirk Weissenborn, Xiaohua Zhai, Thomas Unterthiner, Mostafa Dehghani, Matthias Minderer, Georg Heigold, Sylvain Gelly, Jakob Uszkoreit and Neil Houlsby. “An Image is Worth 16x16 Words: Transformers for Image Recognition at Scale.” ArXiv abs/2010.11929 (2021)
> 2. Touvron, Hugo, Matthieu Cord, Matthijs Douze, Francisco Massa, Alexandre Sablayrolles and Herv'e J'egou. “Training data-efficient image transformers & distillation through attention.” ICML (2021)
> 3. Felzenszwalb, Pedro F., Ross B. Girshick, David A. McAllester and Deva Ramanan. “Object Detection with Discriminatively Trained Part Based Models.” IEEE Transactions on Pattern Analysis and Machine Intelligence 32 (2009): 1627-1645.
> 4. Reddy, M. Vaishnavi, Andrzej Banburski, Nishka Pant and Tomaso A. Poggio. “Biologically Inspired Mechanisms for Adversarial Robustness.” ArXiv abs/2006.16427 (2020)
> 5. Kiritani, Taro and Koji Ono. “Recurrent Attention Model with Log-Polar Mapping is Robust against Adversarial Attacks.”
> 6. Luo, Yan, Xavier Boix, Gemma Roig, Tomaso A. Poggio and Qi Zhao. “Foveation-based Mechanisms Alleviate Adversarial Examples.”
> 7. Freeman, Jeremy and Eero P. Simoncelli. “Metamers of the ventral stream.” Nature neuroscience 14 (2011): 1195 - 1201.

---

> > ### Comment · Reviewer_gFeo · 2021-11-29
> > **Response to authors**
> >
> > I thank the authors for their response and the additional studies to addressing concerns I and other reviewers expressed.  The clarification of vision transformers versus spatial transformers is appreciated and I hope this clarification will avoid similar confusion by some other readers.
> >
> > As to adversarial attacks, I am well familiar with the importance of adversarial robustness for computer vision models in general. This was not my point regarding the lack of motivation behind choosing to evaluate a hybrid foveated vision system based on adversarial robustness. Motivation of the metric should relate to motivation of the model. Can you explain *why* foveation in particular should be expected to provide robustness to adversarial attack?  If foveation is inspired by biological vision systems that have need to foveate, what role does adversarial robustness possibly play in biological vision systems, at least the gradient-based spoofing attacks described here?  It appears that the only motivation behind this line of evaluation is simply that you have 1) proposed a computer vision model of any sort and 2) recent works are testing adversarial robustness of computer vision models. If there is a reason foveation helps with adversarial attacks, which you appear to have demonstrated, can you describe why this might be so or find a way to uncover that mechanism?
> >
> > Direct comparison to the most comparable methods such as Akbas and Eckstein would allow for better assessment of the advantage of the unique aspects of this method, and comparisons to recent deep CNNs such as EfficientNet disappoint. However, I thank the authors for the ablation studies, visualizations and comparisons to random foveations provided in the appendix. These provide interesting exploration of the model that help to understand its true strengths, with some interesting and non-intuitive results. It appears that the fixation prediction provides improved classification performance over random, illustrating that the attentional model is doing something useful. In a somewhat surprising result, the predictions are not better (and actually worse) than random for adversarial robustness. This may reveal something about the utility of the foveation model for adversarial robustness that is not tied to predictive attention, but rather to disrupting the mechanism of the adversarial attack by reducing model predictability. I hope the authors found it intriguing that the stopping criteria is inversely rather than positively correlated to a complexity measure of the object. I agree with the authors that the primary factor in the success of the attentional fixations may be less related to the complexity of a given object, but rather in how efficiently it can find a potentially small object in the background. These studies contribute significantly to understanding the mechanisms at work and make the paper more interesting and informative.

---

> > > ### Author Response · Authors · 2021-11-30
> > > **Response_2 to reviewer gFeo**
> > >
> > > Thanks for the additional comments.
> > >
> > > Adversarial perturbations which do not affect humans seem to affect neural networks detrimentally. A biologically-inspired network might, in principle, acquire adversarial robustness by incorporating the features of the human visual system. As the reviewer states, we show that foveated processing does lead to more robustness to adversarial attacks.  The mechanism is likely related partly to a foveated model's multiplicity of feature representations for an image class related to the various alignments of the fovea and the image.  In addition, the variation in saccade exploration across images also provides additional variability in how the models map the image to foveated features and might also make it more robust to adversarial attacks. As the reviewer notes, the random saccade condition makes the model the least predictable and thus more robust to adversarial attacks. However, for some tasks the random saccade might lead to lower classification accuracy than saccade sequences based on self-attention.
> > >
> > >  See also Reddy et al. ([1]) that showed that two features resulting from foveation, non-uniform sampling, and presence of varying receptive field sizes with eccentricity contribute toward robustness against small adversarial perturbations. Also, Luo et al. ([2]) show that foveation improves the accuracy of CNN under adversarial attacks by using the scale and position invariance of the object in convolution neural networks.
> > >
> > >
> > > References:
> > >
> > > [1] Reddy, M. Vaishnavi, Andrzej Banburski, Nishka Pant and Tomaso A. Poggio. "Biologically Inspired Mechanisms for Adversarial Robustness." ArXiv abs/2006.16427 (2020): n. pag.
> > >
> > > [2] Luo, Yan, Xavier Boix, Gemma Roig, Tomaso A. Poggio and Qi Zhao. "Foveation-based Mechanisms Alleviate Adversarial Examples." ArXiv abs/1511.06292 (2015): n. pag.

---

### Official Review · Reviewer_EWqw · 2021-11-02

**Correctness:** 4
**Technical Novelty And Significance:** 4
**Empirical Novelty And Significance:** 3
**Recommendation:** 6
**Confidence:** 4

**Main Review:**

The method presents a unique idea of computer vision tasks based on the human perceptional system. Since humans archive vision processings with limited computing resources, employing the idea may work for computing tasks with the limited resource, thus, the idea proposed in this paper may give an impact on the community. However, I still wonder about the effectiveness due to the lack of comparison with existing methods.

- Performance of object recognition:
The performance of ImageNet task is far from SOTA. I understand the main claim of the paper is introducing a new scheme that increases the performance of the existing algorithm, but if so, the question is why the paper used Deit-Tiny for only the reference. It is highly desired to combine other methods as well as the proposed foveated scheme.

- Attention model:
Discussion (and hopefully experimental validation) need about the difference between the attention model and the proposed gaze control.


- Foveated transformer:
I wonder which is better between the proposed foveated transformer v.s. dilated convolution since they have similar computational models. Rather, dilated convolution is much similar to the human vision for its gradient-density nature.

**Summary Of The Paper:**

The paper shows a method of Foveated Transformer motivated by a human foveal vision where the spatial resolution varies depending on the focused point. Namely, the method mainly consists of the spatial attention model that controls the gaze point and spatial-varying convolutional filter (dense pooling in center and sparse pooling in peripheral). The former simulates human gaze control and the latter mimics the human retina. The experimental validation is performed by the object recognition task (ImageNet) and adversarial attack tasks and performed better than the Deit-Tiny model that uses a similar number of parameters.

**Summary Of The Review:**

The method presents an interesting and unique approach that is motivated by human vision perception system including gaze and attention control as well as the foveated vision. However, experimental validation is limited, and thus difficult to understand the pure effectiveness. I think other application scenarios such as real-time video recognition may be suit for the scheme than the object recognition tasks for its low-cost computing nature.

---

> ### Author Response · Authors · 2021-11-23
> **Response to reviewer EWqw**
>
> Thanks for your review.
>
> - Regarding ablation studies, please refer to Appendix 6.1. We also implemented a pure-transformer based model instead of a hybrid model as presented in Appendix 6.1.2.
>
>
> - We illustrate the effectiveness of our attention model using an auxiliary task in Appendix 6.1
>
>
>
> - Dilated convolutions do have the exciting properties of capturing context and having large receptive fields. However, they lack some unique aspects of foveated vision, such as varying visual resolution and generating fixation-dependent pooling. In addition, since transformers are better suited for working with pooled features (as pooling loses the 2D structure of the input feature map) and does not contain convolutions, our approach becomes more desirable than dilated convolutions.

---

> > ### Comment · Reviewer_EWqw · 2021-12-02
> > **Thank you for the comment.**
> >
> > Thank you for the comment. I have read your respose and experimental results and would keep the current evaluation since the paper shows a new concept in image recognition, while may have some discussions regarding the perceptional difference between image recognition v.s real scene, that includes shooter (central) biases and/or field of view (of images and real environment).

---

### Official Review · Reviewer_czVG · 2021-11-02

**Correctness:** 3
**Technical Novelty And Significance:** 2
**Empirical Novelty And Significance:** Not applicable
**Recommendation:** 3
**Confidence:** 4

**Main Review:**

[Paper strength]
- The method is well-motivated. The fovea in computer vision can save the computation, improve the robustness of the prediction, and improve the model performance.

- The proposed method achieves better performance than baselines in the adversarial robustness experiments. The author shows the results for the robustness experiments in table 2 that the proposed method achieves better performance than baselines. The author then shows the contributions of the foveated natural of the proposed model in figure 4 for the adversarial robustness.

- The proposed architecture make sense to me. The proposed method uses the foveation module to extract coarse and fine information from the feature map, and then change the fovea locations with the fixation sequence prediction, and transformer in the middle to extract feature with attention. Essentially, there are three components of attention mechanisms: foveation module, sequential fixation, and transformer.

- The author shows the proposed method can run faster than other baselines methods while maintaining a similar performance in terms of the accuracy of the classification tasks.

- The author provides detailed information about the proposed network architecture that is possible for re-implementation.

[Paper weakness]
- The main weakness of this paper is the lack of technical novelty. Compared with the previous DeiT-Tiny, the main contribution of this paper is the foveation module. However, the foveation module is very much like the dilated convolutions [a] and the pyramid of dilated convolutions [b], except here the filter/kernel is average pooling. If the author wants to claim novelty over the family of dilated convolutions, they should provide more discussions, analysis and experiments to show the advantage of the proposed method. Current, I cannot judge from the paper.
The sequential fixation is the recurrent CNN, and there is no other technical contribution as far as I can see.
Essentially, the proposed method is a combination of recurrent CNN, transformer, and dilated convolutions. If the author would like to claim this combination itself is the contribution, there should be more ablation experiments on the contribution of each component.

    [a] Multi-scale context aggregation by dilated convolutions. ICLR 2016.

    [b] ESPNet: Efficient Spatial Pyramid of Dilated Convolutions for Semantic Segmentation. ECCV 2018

- It is not clear to me how to train the sequential fixations. Since the fixation point is a spatial location, directly retrieving it from the attention weights map in the paper is not differentiable. I am confused about how the author does the backpropagation, i.e. how to judge one fixation position is better than another one? As far as I know, this is hard attention implemented by recurrent neural networks output the location, which usually can be solved with spatial transformer[c], LSTM, or other reinforcement learning loss. The task itself is not easy to train.

    [c] Spatial transformer networks. NeurIPS 2015.

- There is no significant performance improvement. As shown in Tables 1 and 2, there is no improvement from the proposed method over other baselines in terms of the accuracy of the image classification task. Although the proposed method can run faster than baselines, I do not think the running speed performance is superior since it is actually slower than the ResNet-18 in table 2.

- The initial fixation should start at the centre of the image. I can see from figure 4 that a random initial fixation point is helpful for the proposed method since it can have multiple choices to choose the ideal locations. However, it is unfair for the full-resolution baseline to have a random initial location since it cannot achieve good performance if the random location is not ideal.

- It is not clear to me how much of the class token could be helpful for the final performance, or does it matter at all?

- Equation 2 seems like the loss is averaged after the classification layer, which is a conflict with figure 1 that the averaging happens before the classification layer. Please ignore this comment if I am wrong.

**Summary Of The Paper:**

This paper proposes a called Foveater which uses a foveated module to extract the information from the feature map with different levels of details and different locations. This proposed method has the architecture that makes sense for the image classification task. However, I found it is difficult to find significant technical novelty in this paper.

**Summary Of The Review:**

I agree with the author that the foveation module could be helpful for the computer vision task. However, it is hard for me to see the technical novelty of the proposed method. Therefore, I lean to reject this paper.

---

> ### Author Response · Authors · 2021-11-23
> **Response to reviewer czVG**
>
> Thanks for your review.
>
> - Regarding the ablation experiments,
>
> We included ablation experiments in Appendix 6.1 illustrating the importance of the attention mechanism, contribution of the CNN backbone, and the importance of the periphery rings.
>
>
> - Regarding the attention weights,
>
> Yes, we agree that a loss function that applied directly to the attention weights would be a more desirable approach, as done in the cases of hard-attention or soft-attention networks. In this work, we follow an indirect approach where the attention weights are applied to the pooled feature vectors to generate the class vector, and the network is optimized to learn to classify based on the class vector. We are forcing the network to learn the attention weights for various resolutions, i.e., pooling sizes. This helps us navigate through the image based on the attention weights of a lower resolution pooled vector. We will explore more direct attention optimization methods in future work. Please also check the Appendix for an auxiliary task where we illustrate the effectiveness of our attention mechanism by comparing the performance of the random fixations to guided fixations.
>
>
> - We agree that the foveated model can get an additional advantage due to the randomness of the initial fixation location. We included the Foveated-Guided-CT in Figure 4 to address it where the initial fixation is fixed to be at the center.
>
>
> - Regarding the class token,
>
> The class token is aggregating the information from all the feature vectors and is finally used for classification. It is possible to train the model without the class token vector, like the object detection DETR ([1]) model, but as our model is based on DeiT ([2]) architecture, we followed their usage of the class token.
>
>
> - Regarding Equation 2,
>
> Averaging before the classification layer is the average of class token vectors from all fixations. And the addition in the loss function is to train the model to maximize its performance for all the fixation conditions simultaneously.
>
>
>
>
>
>
> - References:
>
> 1. Carion, Nicolas, Francisco Massa, Gabriel Synnaeve, Nicolas Usunier, Alexander Kirillov and Sergey Zagoruyko. “End-to-End Object Detection with Transformers.” ArXiv abs/2005.12872 (2020): n. pag.
> 2. Touvron, Hugo, Matthieu Cord, Matthijs Douze, Francisco Massa, Alexandre Sablayrolles and Herv'e J'egou. “Training data-efficient image transformers & distillation through attention.” ICML (2021).

---

> > ### Comment · Reviewer_czVG · 2021-11-25
> > **Thank you for the comments. I am still confused about technical novelty part.**
> >
> > I thank the author for their discussion. Now I understand Equation 2 and how the sequential fixation (attention weights) is trained. I have the following concerns after reading the comment from other reviewers and the discussion provided by the author. I hope the author could address my following concerns.
> >
> > **Technical novelty**. Since the reviewer NG64 thinks this paper proposes quite a novel approach while I disagree with the reviewer, I am eager to know the key technical contribution of this paper from the author. I hope the author could directly argue with me on them. Especially:
> > - Comparing with DeiT-Tiny, what is the technical contribution of the proposed method? Is it the foveation module?
> > - Comparing with previous dilated convolutions, what is the technical contribution of the proposed foveation module? The reviewer EWqw has a similar concern. If the author thinks there is an improvement over the dilated convolutions, I think they should provide a comparison with the dilated convolutions methods.
> > - If there is no strong technical contribution for the individual module, does the author argue the combination of recurrent CNN, transformer, and dilated convolutions is the key technical contribution? In this case, I think the contribution actually is limited since the proposed method does not outperform other SOTA methods.
> >
> > **Attention weights**. Thanks for the author's further explanation, now I understand the attention weights are trained with the classification backpropagation. It essentially assumes the higher activation the higher attention which holds the same assumption with the hard-attention mechanism. I do not ask the author to explore more direct attention optimization since it is not a trival task.
> >
> > **Performance**. As we can see in Tab. 2, the proposed method is not better than the previous SOTA methods, not even the ResNet-18 (69.8 vs 69.9 is not significant). I hope the author could put a strong stance to argue why this paper should be accepted even the proposed does not achieve the best performance.

---

> > > ### Author Response · Authors · 2021-11-29
> > > **Response_2 to reviewer czVG**
> > >
> > > Thanks for your follow-up questions.
> > >
> > >
> > > Sequential processing is the main difference between our model and the Deit-Tiny model.
> > >
> > > The Foveation module working along with the attention mechanism is making the sequential processing of the image possible.
> > >
> > > Some of the advantages of our model compared to dilated convolutions,
> > > 1. model gets the ability to distribute different amounts of resources based on the image’s difficulty.
> > > 2. Resulting network is more robust to adversarial attacks
> > > 3. The model uses limited resources at each fixation /step ([1]), even when larger input size is used.
> > > 4. Computation savings when compared to the full resolution counterpart
> > > 5. A Foveation module is a plug-in module that can be used in hybrid architectures or pure transformer networks, whereas dilated convolutions can only work CNN architectures.
> > >
> > > References:
> > > Mnih, Volodymyr, Nicolas Manfred Otto Heess, Alex Graves and Koray Kavukcuoglu. “Recurrent Models of Visual Attention.” NIPS (2014).

---

> > > > ### Comment · Reviewer_czVG · 2021-12-02
> > > > **I would like to remain my current rate**
> > > >
> > > > I think the author's feedback and useful discussion. I also read comments and discussions of other reviewers, and I want to keep my current rate. It is unfortunate that reviewers do not agree with each other and have large discrepancies in scores.
> > > >
> > > > The main reason for my decision is it is really struggling for me to appreciate the technical contribution of this paper. If the sequential processing is the main improvement over the most related work (Deit-Tiny model) while the author emphasizes the contribution is "foveated transformer" which makes me confused. The author has many arguments that the proposed method is better than dilated convolutions family group, which I sincerely need ablation study to be convinced rather than text arguments. From my point of view, the proposed method is a combination of recurrent CNN (or sequential processing), transformer, and dilated convolutions. Therefore, I think this paper is not good enough for the ICLR.

---

### Official Review · Reviewer_NG64 · 2021-11-03

**Correctness:** 3
**Technical Novelty And Significance:** 4
**Empirical Novelty And Significance:** 3
**Recommendation:** 8
**Confidence:** 4

**Main Review:**

Strengths:

- Overall, I found this paper both an interesting read and quite a novel approach. It is great to see practical sequential models being developed, and the inclusion of robustness evaluation via adversarial attacks was very helpful in bolstering the paper's claims.

- I appreciate the effort made to make the model comparisons more fair (i.e. based on a similar number of parameters).

- The components of the model, training steps, and experiments were clearly described.

Weaknesses and Questions:

- Why restrict the adversarial attacks to only the transformer component? We know adversarial attacks affect CNNs, too, so I was unclear on the rationale for this design choice. Also, since Deit-Ti is a purely transformer-based method, I found it unclear if this meant that the comparison of the adversarial attack on Deit-Ti against the two hybrid methods proposed in the paper was fully equivalent.

- What is Deit-Ti (distilled)? I could not find an explanation for this model in the text, and it only appears in Table 2.

- Why is the Dynamic Stop version of the foveated model not included in the adversarial robustness investigation?

- In Table 1, one column uses "PC" as a column head, but I couldn't find an explanation of what this stood for. It is possible I missed it, though. I am assuming from context it is performance.

- In Section 3.2, I found it somewhat unclear to follow. Are there two 50% decay mechanisms (1. the attention weights of the previous confidence map, 2. the IOR map of previously fixated locations)?

- The line directly below equation (2) has a typo: "fixatiothe n"

- In Section 2: Sequential processing, the way the three main advantages are interspersed with the rest of the paragraph text is a little hard to follow. Also, an additional advantage of sequential processing with eye movements is articulated by Tsotsos (2011) A Computational Perspective on Visual Attention, which is to compensate for the "Boundary Problem" introduced by the loss of a convolution kernel half-width of fully defined output for each layer of convolutional filtering. The relevance of this issue to CNNs was demonstrated by Alsallakh et al., Mind the Pad - CNNs Can Develop Blind Spots, ICLR 2021.

- In Section 2: Computational models of categorization and eye movements, it is mentioned that saliency-based models do not incorporate foveal vision, but Wloka et al., Active Fixation Control to Predict Saccade Sequences, CVPR 2018 provides a foveated model of eye movements based on an underlying saliency map.

**Summary Of The Paper:**

The paper introduces a fixation-based method based on a hybrid transformer-CNN architecture, which demonstrates many of the strengths of sequential processing, namely computational efficiency and adversarial robustness. This model largely outperforms a comparative full resolution model, and has a number of attractive behavioural qualities.

**Summary Of The Review:**

This is an overall strong paper. It proposes an interesting new approach, and does a good job of exploring the behaviour of this approach with respect to alternative architectures. The weaknesses identified are largely minor, points of clarity, or easily fixable.

---

> ### Author Response · Authors · 2021-11-23
> **Response to reviewer NG64**
>
> Thanks for your review.
>
>
> - Regarding the design choice for adversarial robustness,
>
> Yes, that is correct. In order to understand the effect of the foveation module, we compared the transformer layers that follow the foveation module in Foveated version to the equivalent part in the Full-resolution version. We also noticed more gains in adversarial robustness when the attack is applied at the foveation module, thereby restricting what the adversarial attack can see. We will explore more ways to quantify the contributions of the foveation module in the future.
>
> - Regarding DeiT-Ti (distilled),
>
> it stands for the distillation method used by the DeiT ([1]) authors, where there is a convolutional network acting as a teacher and helps to train the student transformer network.
>
>
> - Since the dynamic stop model’s intention is to reduce the number of computations, we did not include the Adversarial robustness results. Under FGSM attack with an epsilon of 0.01, Top-1 accuracy of Foveated model with Dynamic-stop is 56.4% with 3.8 fixations per image. Compared to the same attack on the same model without Dynamic-stop, there is a decrease of ~2.8% in accuracy and an increase of one fixation per image.
>
>
> - Regarding “PC”, We used this acronym for Top-1 accuracy by mistake, updated text.
>
>
> - Regarding decay mechanisms,
>
> Yes, that is correct. There are two 50% decay mechanisms, (1). Decay for attention weights that are mapped backed onto 14x14 map, (2). The IOR map consists of inhibition in the fovea region.
>
> - Fixed the minor errors and added the additional references.
>
>
> - References:
> 1. Touvron, Hugo, Matthieu Cord, Matthijs Douze, Francisco Massa, Alexandre Sablayrolles and Herv'e J'egou. “Training data-efficient image transformers & distillation through attention.” ICML (2021).

---

> > ### Comment · Reviewer_NG64 · 2021-12-02
> > **Thank you for your updates**
> >
> > I have read the authors' updates and the other reviews. I would like to thank the authors for engaging thoughtfully with the critiques and updating the text to be clearer.
> >
> > Regarding the different assessments between reviewers, while I agree that the performance demonstrated in this paper is not exceptional, that is not the only dimension through which progress can be made. I stand by my initial assessment that this a good paper and useful to have published at a top-tier venue like ICLR for the following reasons:
> >
> > Iterative foveal vision is quite different from standard approaches and is challenging to implement in a useful manner. Achieving performance on par with more traditional approaches on a dataset which does not inherently showcase many of the actual performance benefits which are provided by a foveal mechanism (e.g. greater robustness for object detection when objects are not well centered and consistent resource requirements to flexibly handle different sized inputs and different spatial scales of the input) is therefore a strong technical achievement. While I think the authors may have been able to make the strengths of the foveal mechanism more readily apparent by including additional experiments which explicitly showcase these aspects of performance, the investigation of adversarial robustness serves as an interesting dimension to explore as well, and leads to some intriguing questions for future work, particularly with regard to the difference between random sampled fixations versus guided fixations (a finding highlighted as interesting by other reviewers).

---

### Author Response · Authors · 2021-11-23
**Common response**

We thank all the reviewers for their reviews. We added individual replies to all the reviews. On a common note, changes in the main paper are in red for the convenience of the reviewers. New sections are added in the Appendix.

---

### Decision · Program_Chairs · 2022-01-20

**Decision:**

Reject

**Comment:**

This paper introduces an architecture that uses pooling regions and
eye movements to sequentially build up an object representation.  A
confidence threshold is used to allow recognition in less time for
easier images.

There was a lot of disagreement on this paper.  Those in favor argued
that it is a worthy endeavor to explore new biologically motivated
architectures and foveated eye movements are an important aspect of
human vision that is worth exploring for computer vision.  Another pro
was the improved robustness to some adversarial attacks.  Those
arguing for not accepting the paper, argued that classification
performance is not improved over SOTA and that more ablation studies
should be done to better understand the role and importance of the
various aspects of the model and how they differ from other
architectural designs with dilated convolutions instead of the
foveation module.

I agree that more ablation studies would be useful to better
understand the role of the different model components. While I
feel that this novel sequential processing algorithm is worth publishing to
increase activity in this area, I feel it would be best received after further
studies help clarify the importance of different aspects of the model.
I recommend resubmission after further analysis.